# Usefulness of the Urine Methylation Test (Bladder EpiCheck^®^) in Follow-Up Patients with Non-Muscle Invasive Bladder Cancer and Cytological Diagnosis of Atypical Urothelial Cells—An Institutional Study

**DOI:** 10.3390/jcm11133855

**Published:** 2022-07-03

**Authors:** Karla B. Peña, Francesc Riu, Anna Hernandez, Carmen Guilarte, Joan Badia, David Parada

**Affiliations:** 1Molecular Pathology Unit, Department of Pathology, Hospital Universitari de Sant Joan, 43204 Reus, Spain; karlabeatriz.pena@salutsantjoan.cat (K.B.P.); francesc.riu@salutsantjoan.cat (F.R.); anna.hernandez@salutsantjoan.cat (A.H.); carmen.guilarte@salutsantjoan.cat (C.G.); 2Institut d’Investigació Sanitària Pere Virgili, 43204 Reus, Spain; joan.badia@iispv.cat; 3Facultat de Medicina i Ciències de la Salut, Universitat Rovira i Virgili, 43007 Reus, Spain

**Keywords:** methylation, DNA, urothelial, cancer, bladder, atypia, cytology

## Abstract

Urothelial bladder cancer is a heterogeneous disease and one of the most common cancers worldwide. Bladder cancer ranges from low-grade tumors that recur and require long-term invasive surveillance to high-grade tumors with high mortality. After the initial contemporary treatment in non-muscle invasive bladder cancer, recurrence and progression rates remain high. Follow-up of these patients involves the use of cystoscopies, cytology, and imaging of the upper urinary tract in selected patients. However, in this context, both cystoscopy and cytology have limitations. In the follow-up of bladder cancer, the finding of urothelial cells with abnormal cytological characteristics is common. The main objective of our study was to evaluate the usefulness of a urine DNA methylation test in patients with urothelial bladder cancer under follow-up and a cytological finding of urothelial cell atypia. In addition, we analyzed the relationship between the urine DNA methylation test, urine cytology, and subsequent cystoscopy study. It was a prospective and descriptive cohort study conducted on patients presenting with non-muscle invasive urothelial carcinoma between 1 January 2018 and 31 May 2022. A voided urine sample and a DNA methylation test was extracted from each patient. A total of 70 patients, 58 male and 12 female, with a median age of 70.03 years were studied. High-grade urothelial carcinoma was the main histopathological diagnosis. Of the cytologies, 41.46% were cataloged as atypical urothelial cells. The DNA methylation test was positive in 17 urine samples, 51 were negative and 2 were invalid. We demonstrated the usefulness of a DNA methylation test in the follow-up of patients diagnosed with urothelial carcinoma. The methylation test also helps to diagnose urothelial cell atypia.

## 1. Introduction

Bladder cancer is the tenth most common cancer worldwide, with approximately 573,000 new cases and 213,000 deaths [1,2]. In men, the respective incidence and mortality rates are 9.5 and 3.3 per 100,000 more common than in women [1], and it is the sixth most common cancer and the ninth leading cause of cancer death. In Southern Europe (Greece, Spain and Italy), Western Europe (Belgium and The Netherlands) and North America, the incidence rates are highest for both sexes [1,2].

Bladder cancer is one of the cancers with the longest lifespans and highest costs due to the high rate of recurrence and the need for continuous monitoring [3,4]. Currently, both cystoscopy and urinary cytology are the most common methods of diagnosis in both the detection and the follow-up of malignant urothelial neoplasms [5,6,7,8]. However, both methods have advantages and disadvantages. Cystoscopy is an invasive method and causes significant discomfort to patients [5,6,7,8], while urinary cytology is non-invasive and effective in diagnosing high-grade (HG) urothelial cancers but has a sensitivity between 11% and 17% to low-grade (LG) urothelial lesions, which are the most common lesions of the bladder [5,6,7,8]. Additionally, the consistency and precision of the cytomorphological evaluation may undergo alterations as a consequence of the treatment given for primary or recurrent urothelial neoplasms of the urinary bladder and inflammatory states, which makes a definitive diagnosis difficult [9,10,11,12].

Urothelial cells with abnormal cytological characteristics that do not meet the criteria for malignancy are commonly found in the daily practice of urinary cytopathology. In this atypia of urothelial cells, the clinical−cytological correlation is not adequate, and its diagnostic approach remains difficult [13,14,15]. Various reasons have been given for this: for example, borderline neoplastic urothelial morphological alterations, changes associated with benign processes, such as inflammation, lithiasis, or the effect of local treatment, and even poorly fixed samples [13,14,15,16,17,18]. Therefore, urinary biomarkers need to be found that can be used in patients with atypical urothelial cells so that they can be studied appropriately. The main objective of our study was to evaluate the usefulness of a urine DNA methylation test in patients with urothelial bladder cancer under follow-up and a cytological finding of urothelial cell atypia. In addition, we analyzed the relationship between the urine DNA methylation test, urine cytology, and subsequent cystoscopy study.

## 2. Materials and Methods

### 2.1. Study Design and Patient Cohort

This is a prospective and descriptive cohort study conducted on patients with non-muscle invasive urothelial carcinoma under oncological and urological follow-up at the Urology and Medical Oncology Service of the South Catalonia Oncology Institute (Hospital Universitari de Sant Joan, Reus, Spain), between 1 January 2018 and 31 March 2022. Patients with painless hematuria and Paris system category III, IV, and V, with no histopathological diagnosis were also included (Figure 1). We studied urine samples that 70 patients had submitted to the molecular pathology unit of our pathology department. The patients’ clinical data were extracted from medical records and the study was approved by the Institutional Review Board (IISPV). Two urine samples were obtained from each patient simultaneously, one of which was processed for cytological study and the other for DNA methylation study. At the time of evaluating the results of both the cytology study and the methylation test, none of the investigators responsible for the analyses were aware of the results of the tests.

### 2.2. Urine Cytology Study

Urine samples were routinely processed with liquid-based cytology using the Thin Prep 5000 TM method (Hologic Co., Marlborough, MA, USA). All the sample material was fixed with the hemolytic and preservative solution Cytolyt^TM^ and spun at 3000 rpm for 5 min. The sediment was then transferred to 20 mL of PreservCyt solution, kept for 15 min at room temperature, and processed with a T5000 automated processor in accordance with the manufacturer’s recommendations. One slide was obtained for each sample and fixed in 95% ethanol. The slide was stained with Papanicolaou. All samples were evaluated according to the Paris System to report urine cytology. The diagnostic categories were: category I: insufficient material for diagnosis; category II: negative for high-grade urothelial carcinoma; category III: atypical urothelial cells; category IV: suspicious for high-grade urothelial carcinoma; and category V: high-grade urothelial carcinoma.

### 2.3. DNA Methylation Study (Bladder EpiCheck^®^ Test)

The Bladder EpiCheck^®^ is an in vitro diagnostic test for the detection of DNA methylation patterns in urine that are associated with bladder cancer. A cell pellet was created from every urine sample for the Bladder EpiCheck^®^ test (Nucleix, Rehovot, Israel). The urine sample was centrifuged twice at 1000× *g* for 10 min at room temperature. DNA was extracted from the cell pellet using the Bladder EpiCheck^®^ DNA extraction kit. The extracted DNA was digested using a methylation sensitive restriction enzyme, which cleaves DNA at its recognition sequence if it is unmethylated. The quantitative real-time polymerase chain reaction (qRT-PCR) amplification was performed using Rotor-Gene Q. The samples were prepared for the PCR assay using the Bladder EpiCheck^®^ test kit, and the results were analyzed using the Bladder EpiCheck^®^ software. For samples that pass the internal control validation, the software calculates an EpiScore (between 0 and 100) which represents the overall methylation level of the sample on the panel of biomarkers. The test cut-off is an EpiScore of 60. An EpiScore ≥ 60 indicates a high probability of bladder cancer (positive), and a score < 60 indicates a high probability of no bladder cancer or that the cancer is still in remission (negative). An invalid result indicates the test should be repeated.

### 2.4. Statistical Analysis

Statistical analyses were carried out in “R” (version 4.2.0) using “stats” library (version 4.2.0). Graphs were elaborated in “R” using “ggpubr” and “ggplot2” packages (version 0.4.0 and 3.3.6). Bladder cancer was diagnosed by a pathologist and set as the reference standard against which both urine cytology and bladder epicheck were compared to assess the sensitivity, specificity, positive predictive value and negative predictive. Cytology results were considered negative for categories I and II, atypical for category III and positive for categories IV and V. 

## 3. Results

### 3.1. Clinical Findings

Our analysis included 58 male and 12 female patients with a median age of 70.02 years (range 49–91 years). In the group under follow-up for urothelial bladder cancer (59 patients), there were 52 males (88.14%) and seven females (11.86%) with a mean age of 71.34 years (range 50–91 years). The previous histopathological diagnosis was carcinoma in situ in six patients (10.17%), high-grade urothelial carcinoma in 37 patients (62.71%), and low-grade urothelial carcinoma in 16 (27.12%) cases. The follow-up time was one year or less in 14 patients (23.73%), between two and five years in 26 patients (44.07%), between six and ten years in 14 patients (23.73%), and more than ten years in five patients (8.47%). Recurrence was observed in 29 patients (49.15%). Thirty-eight (64.40%) patients received intravesical therapy with BCG (see Table 1).

### 3.2. Cytologic Findings

A total of 82 urinary cytologies from 70 patients were analyzed. Of these, six (7.32%) were category I; 25 (30.49%) were category II; 34 (41.46%) were category III; 16 (19.51%) were category IV; and one (1.22%) was category V. The urothelial carcinoma follow-up group contained 71 of these 82 urinary cytologies. In this group, the main diagnostic category was III (21 cytologies), followed by category II (17 cytologies) and category IV (14 cytologies) (Figure 2). In the group with painless hematuria (11), the main diagnostic category was III (8 cytologies), followed by category IV (2 cytologies) and category II (1 cytology) (see Table 1).

### 3.3. DNA Methylation Test (Bladder EpiCheck^®^ Test) Findings

In the follow-up urothelial carcinoma group, a total of 71 DNA methylation tests were performed. Two urine tests were invalid for the DNA methylation test. Of the remaining samples, 18 (25.35%) were positive and 51 (71.83%) were negative for the DNA methylation test. In the group with painless hematuria, all samples were negative for the DNA methylation test (see Table 1).

### 3.4. Relationship between Urinary Cytological Findings and the DNA Methylation Test (Bladder EpiCheck^®^ Test)

For the patients in the urothelial carcinoma group in diagnosis category I (6), the DNA methylation test was negative in five cases and invalid in one. For diagnostic category II (17), the DNA methylation test was negative in 15 cases and positive in two. For diagnostic category III (21), consisting of atypical urothelial cells (AUC), the test was negative in 16 cases and positive in five. In cytological categories IV and V (15), the DNA methylation test was negative in one case, positive in 13 cases and invalid in one. In the painless hematuria group, the DNA methylation test was negative for all the urinary cytologies diagnosed as category II (1), III (8), and IV (2).

The overall sensitivity rates for the DNA methylation test and cytology were 91.67% and 90%, respectively, while the overall specificity rates were 91.89% and 88.89%, respectively. The overall PPV was 81.82% for cytology and 78.57% for the DNA methylation test; the overall NPV was 94.12% for cytology and 97.14% for the DNA methylation test (Table 2, Figure 3).

The high-grade urothelial carcinoma sensitivity rates for the DNA methylation test and cytology were 85.71% and 90.91%, respectively, while the specificity rates were 92.31% and 83.33%, respectively. The PPV was 62.50% for the cytology and 70.57% for the DNA methylation test; the NPV was 96.77% for the cytology and 96.77% for the DNA methylation test (Table 3).

## 4. Discussion

Bladder cancer is mainly diagnosed in the third decade of life [19] and, at the time of diagnosis, patients are in a treatable stage with a long life expectancy but require long periods of surveillance, follow-up, and treatment of recurrences and complications [20,21]. In the urothelial carcinoma group in our study, there was a predominance of males (88.24%), with a mean age of 77.9 years, and more than 80% of the patients had a follow-up time of between one and ten years. [1,2,3,4,20,21]. This high recurrence rate of 47.45% evidenced in our study is similar to that described in other publications, in which the recurrence rate can reach 52% at five years, implying a significant prevalence of non-muscle invasive bladder carcinoma (NMIBC). All these data demonstrate a significant burden for the patient and physician, and a high economic impact related to patient care [20,21].

The category of NMIBC includes different types of lesions, such as non-invasive neoplasms, which are those that invade the subepithelial connective tissue and carcinoma in situ. These tumors are frequently treated with different therapeutic options depending on variables such as the histopathological grade and the stage within the clinical context [21]. However, the treatment has the capacity to produce various morphological and cytological alterations that increase the diagnostic difficulty in the follow-up of these patients. For example, it has been described that treatment with immunomodulators such as BCG produces reactive urothelial atypia, in addition to urothelial denudation, granulomatous inflammation, eosinophilic cystitis, and persistence of carcinoma in situ in von Brunn nests [9,10]. In our study, the presence of urothelial atypia was found in 35.59%. In actuality, the rate of notification of urothelial cell atypia ranges between 2 and 31% [22,23,24,25]. This greater number of urine cytologies with urothelial atypia in our study could be explained by the changes induced by the treatment administered in our patients.

Currently there are different new urinary biomarkers based on genetic or epigenetic abnormalities that are common in bladder cancer, such as aberrant DNA methylation and non-coding RNA [26,27,28,29,30,31,32,33]. Several of these tests have demonstrated their usefulness in the follow-up of NMIVT patients with very high negative predictive values (NPV) for recurrences of NMIBC (high-grade), which raises the possibility of adapting them to the follow-up of patients [26,27,28,29,30,31]. In our study, we evaluated the capacity of the DNA methylation test in patients with a cytological diagnosis of urothelial atypia and a cystoscopy study. Thus, the methylation test showed that most of the urothelial atypia cases were negative for the methylation test as well as for the cystoscopy study, which represents a diagnostic aid in this diagnostic category and could be used safely in these patients. In addition, molecular markers can help improve the interpretation capacity of other diagnostic tests, such as urine cytology, through positive feedback. However, additional studies are needed to demonstrate the usefulness of the DNA methylation test in the setting of cytological atypia.

Our analysis of the results of urinary cytology samples and the DNA methylation test showed a reasonable relationship between the negative results of the two methods. In general, the NPV was 96.77% for both the cytology and DNA methylation test in high-grade lesions, but when we analyzed all the lesions, the methylation test showed an value of the NPV, arising to 97.14% and cytology arising to 94.12%. These findings confirm the NPV of the DNA methylation test [34,35]. In addition, molecular markers can help improve the interpretation capacity of other diagnostic tests, such as urine cytology, through positive feedback.

Our study has certain limitations including the small sample size and only six patients having more than one sample evaluated by both cytology and DNA methylation testing; most of our sample had the typical limitation of single visit studies. The inclusion of low-grade urothelial carcinomas can modify the usefulness of the methylation test. However, during the molecular evolution of urothelial carcinoma, 20% of low-grade carcinomas can evolve to high-grade, which is why the test may be useful in monitoring them. Currently, the meaning of the variation in bladder EpiScore values and whether it can provide information on the risk of developing urothelial cancer or response to treatment is unknown.

## 5. Conclusions

In conclusion, we demonstrate the usefulness of a DNA methylation test in the follow-up of patients diagnosed with urothelial carcinoma and in patients with painless hematuria. The methylation test also helps in the diagnosis of urothelial cell atypia, which involves savings in subsequent clinical studies. New prospective research is needed to define whether the quantitative value of the DNA methylation test can be used as a prognostic factor to predict response to treatment.

## Figures and Tables

**Figure 1 jcm-11-03855-f001:**
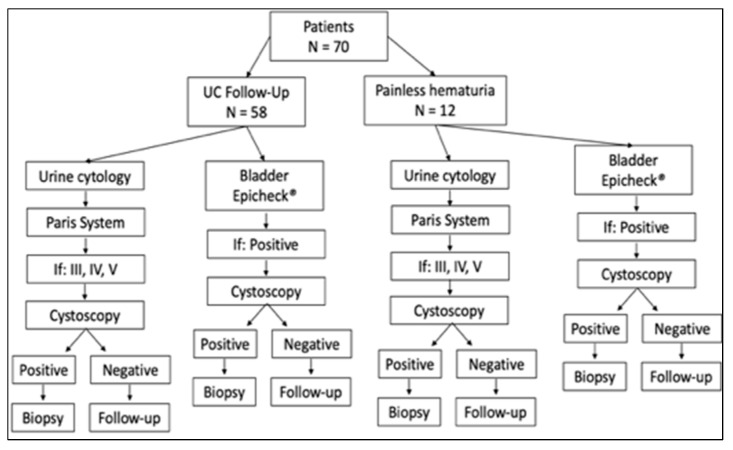
Distribution and workflow for patients included in the present study (N = 70). UC: Urothelial carcinoma.

**Figure 2 jcm-11-03855-f002:**
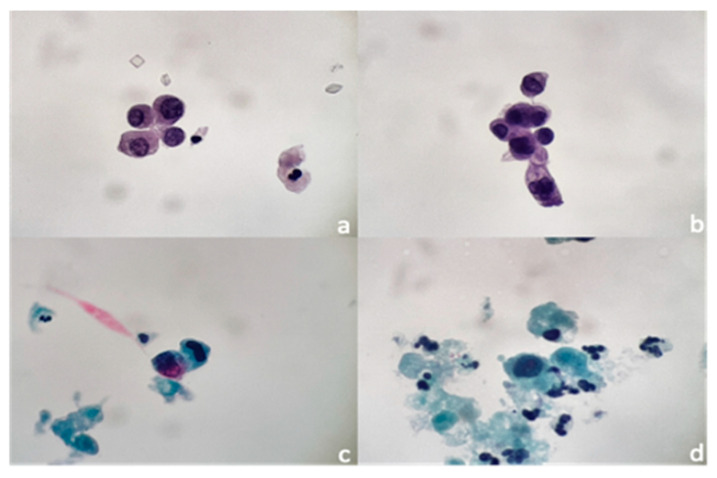
Cytological findings in cases under follow-up for urothelial carcinoma. (**a**) Characteristic finding of atypia urothelial cell. Hyperchromatic nuclei and nuclei/cytoplasm loss. (**b**) Isolated group with evidence of cytologic atypia. Some cytoplasm has a vacuolated aspect with hyperchromatic nuclei. (**c**) Two isolated urothelial cells with marked cytologic atypia suspicious for carcinoma. (**d**) Single urothelial cell with marked loss of nucleus−cytoplasm ratio, nuclear hyperchromatism, and presence of nucleolus. (Papanicolau staining. DA 20× and 40×).

**Figure 3 jcm-11-03855-f003:**
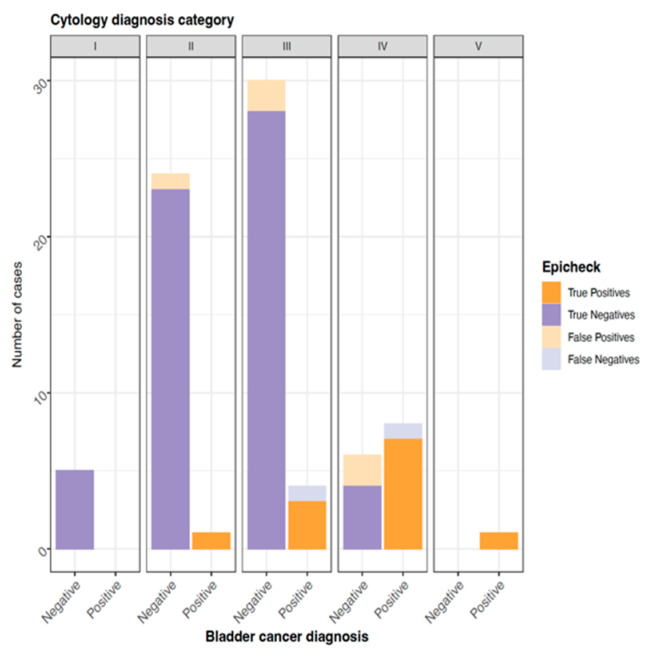
Overall concordance between EpiCheck and bladder diagnosis.

**Table 1 jcm-11-03855-t001:** Clinical characteristics of patients under investigation for DNA methylation test (N = 70).

Histopathological Diagnosis	LGUC	HGUC	CIS	No Cancer
Patients (N)	16	37	6	11
Age (years) (minimum-maximum)	58–82	50–91	53–65	49–76
Gender				
Male	15	31	6	6
Female	1	6	0	5
Cytological diagnosis (PSC)				
I	5	1	0	0
II	5	11	1	1
III	3	14	4	8
IV	3	10	1	2
V	0	1	0	0
DNA methylation test				
Positive	3	12	2	0
Negative	12	24	4	11
Invalid	1	1	0	0
Primary tumor (Bladder)				
Yes	16	34	6	0
No	0	3	0	11
Cystoscopy				
Positive	3	11	2	0
Negative	11	22	2	5
Unrealized	2	4	2	6
Follow-up time (years)				
<1	1	9	4	10
2–5	10	15	1	1
6–10	4	9	1	0
>10	1	4	0	0
Recurrence				
Yes	11	16	2	11
No	5	21	4	0
Treatment				
BCG	4	28	6	0
Mitomycin	1	0	0	0
Chemotherapy	0	2	0	0
Surgery	1	4	0	0
No	10	3	0	11

PSC: Paris System Category.

**Table 2 jcm-11-03855-t002:** Overall sensitivity, specificity, positive predictive value, and negative predictive value of bladder EpiCheck^®^ and cytology in follow-up non-muscle invasive bladder carcinoma.

	Non-Muscle Invasive Carcinoma	
	Bladder EpiCheck^®^	Cytology
Sensitivity	91.67%	90%
Specificity	91.89%	88.89%
PPV	78.57%	81.82%
NPV	97.14%	94.12%

**Table 3 jcm-11-03855-t003:** Sensitivity, specificity, positive predictive value, and negative predictive value of bladder EpiCheck^®^ and cytology in high-grade urothelial carcinoma.

	Non-Muscle Invasive Carcinoma	
	Bladder EpiCheck^®^	Cytology
Sensitivity	85.71%	90.91%
Specificity	92.31%	83.33%
PPV	70.59%	62.50%
NPV	96.77%	96.77%

## Data Availability

All of the data are present in the manuscript.

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
