# Peer review of "Usefulness of the Urine Methylation Test (Bladder EpiCheck®) in Follow-Up Patients with Non-Muscle Invasive Bladder Cancer and Cytological Diagnosis of Atypical Urothelial Cells—An Institutional Study"

_jcm, 2022, doi:10.3390/jcm11133855_

Round 1

Reviewer 1 Report

The authors improved the manuscript accordingly to previous suggestions. Nevertheless, the references in the discussion should be updated. Please enrich your manuscript with https://doi.org/10.1016/j.critrevonc.2022.103577 and https://doi.org/10.1002/cac2.12129

Author Response

Prof. Dr. Massimiliano Creta

Special Issue Editor

Journal Clinical Medicine

Reus, June 26, 2022

Dear Prof. Dr. Massimiliano Creta,

Enclosed please find the manuscript entitled “Usefulness of the Urine Methylation Test (Bladder EpiCheck®) in Follow-up Patients with Non Muscle invasive Bladder Cancer and Cytological Diagnosis of Atypical Urothelial Cells. An Institutional Study”, which we are resubmitting ("This 
manuscript is a resubmission of jcm-1724948) as an original contribution for editorial consideration by your journal. All the authors have made a substantial contribution to the information or material submitted for publication and we have read and approved the final manuscript.  None of us have any direct or indirect commercial financial incentive associated with publishing the article and we have indicated the source of extra-institutional funding, particularly that provided by commercial sources. Finally, the manuscript, or portions thereof, is not under consideration by any other journal or electronic publication, and has not been published and is not under current consideration elsewhere.

We appreciate your time in evaluating our study, as well as that of the reviewers. Below, we point out the changes made by the comments received, as follows:

Reviewer 1:

Dear reviewer, thank you for your valuable comment and time.

1) Point 1: The authors improved the manuscript accordingly to previous suggestions. Nevertheless, the references in the discussion should be updated. Please enrich your manuscript with https://doi.org/10.1016/j.critrevonc.2022.103577 and https://doi.org/10.1002/cac2.12129

 1.1) Response to point 1: Following your recommendation, we have included the following references:

Line : 387-388: [33] Huang HM, Li HX. Tumor heterogeneity and the potential role of liquid biopsy in bladder cancer. Cancer Commun (Lond). 2021 Feb;41(2):91-108. doi: 10.1002/cac2.12129. Epub 2020 Dec 30. PMID: 33377623; PMCID: PMC7896752.

Line : 389-391: [34] Crocetto F, Barone B, Ferro M, Busetto GM, La Civita E, Buonerba C, Di Lorenzo G, Terracciano D, Schalken JA. Liquid biopsy in bladder cancer: State of the art and future perspectives. Crit Rev Oncol Hematol. 2022 Feb;170:103577. doi: 10.1016/j.critrevonc.2022.103577. Epub 2022 Jan 5. PMID: 34999017.

Thank you very much, in advance, for your consideration.

Sincerely,

Dr. David Parada D,

David Parada D, MD, PhD. Unit of Molecular Pathology, Pathology Service. University Hospital of Sant Joan, Faculty of Medicine, IISPV, “Rovira i Virgili” University, Reus, Tarragona, Spain.

Reviewer 2 Report

I would like to congratulate with the authors for providing a revised version of the manuscript. All my comments have been correctly addressed.

Author Response

Prof. Dr. Massimiliano Creta

Special Issue Editor

Journal Clinical Medicine

Reus, June 26, 2022

Dear Prof. Dr. Massimiliano Creta,

Enclosed please find the manuscript entitled “Usefulness of the Urine Methylation Test (Bladder EpiCheck®) in Follow-up Patients with Non Muscle invasive Bladder Cancer and Cytological Diagnosis of Atypical Urothelial Cells. An Institutional Study”, which we are resubmitting ("This 
manuscript is a resubmission of jcm-1724948) as an original contribution for editorial consideration by your journal. All the authors have made a substantial contribution to the information or material submitted for publication and we have read and approved the final manuscript.  None of us have any direct or indirect commercial financial incentive associated with publishing the article and we have indicated the source of extra-institutional funding, particularly that provided by commercial sources. Finally, the manuscript, or portions thereof, is not under consideration by any other journal or electronic publication, and has not been published and is not under current consideration elsewhere.

We appreciate your time in evaluating our study, as well as that of the reviewers. Below, we point out the changes made by the comments received, as follows:

Reviewer 2:

Dear reviewer, thank you for your valuable comment and time.

 Sincerely,

Dr. David Parada

David Parada D, MD, PhD. Unit of Molecular Pathology, Pathology Service. University Hospital of Sant Joan, Faculty of Medicine, IISPV, “Rovira i Virgili” University, Reus, Tarragona, Spain.

This manuscript is a resubmission of an earlier submission. The following is a list of the peer review reports and author responses from that submission.

Round 1

Reviewer 1 Report

Authors tried to test the accuracy of the Bladder EpiCheck test for the diagnosis of urothelial carcinoma in patients with Cytological diagnosis of Atypical Urothelial Cells. I believe the manuscript should not be considered for publication due to several major limits:

-small number of patients (only 27 with Atypical cells)

-heterogeneity of the cohort (previous HG tumors, previous LG tumors vs no previous diagnosis of cancer)

-lack of enough information for the study cohort: how many diagnosis of cancer per patient? Previous intravescical therapies? Number and size of the lesions?

-The greatest limitation is the lack of a reference after the tests? How do we know that either cytology or EpiCheck correctly discovered a tumor? How can we calculate sensitivity, specificity etc.? Without these information, the study is not useful for the introduction of EpiCheck in daily practice.

Author Response

Dear reviewer, thank you for your valuable comment and time. We have collected your comments in order to improve our work and so that it can be assessed for possible publication. Here are the changes made:

1) Point 1: Small number of patients (only 27 with Atypical cells). Heterogeneity of the cohort (previous HG tumors, previous LG tumors vs no previous diagnosis of cancer):

1.1) Response to point 1: During the review we have increased the number of patients included in the study, who are being followed up for non-muscle invasive urothelial carcinoma.

We believe it is necessary to modify the title of the work, since the patients with urothelial carcinoma were non-muscle invasive and therefore in oncological follow-up.

The heterogeneity in the choice of the patients in the study was based on:

  1. a) Patients under follow-up with a previous histopathological diagnosis of in-situ urothelial carcinoma, which are high-grade lesions with a high probability of invasion during their evolution, and their diagnosis would imply histopathological confirmation and therapeutic intervention.
  2. b) Patients under follow-up with high-grade papillary urothelial carcinoma whose diagnosis requires diagnostic and therapeutic interventions.
  3. c) Follow-up patients with low-grade papillary urothelial carcinoma. At this point, we take into consideration the evolution of this type of neoplasm, since approximately 20% of these neoplasms have been described as evolving into high-grade urothelial carcinomas, which requires histopathological confirmation and therapeutic intervention.
  4. d) Patients without a diagnosis of urothelial carcinoma were included, since in our hospital, prior to the validation study for the methylation test, it was decided to use the test as part of the initial diagnosis of urothelial carcinoma, taking into account the negative predictive value Of the test.

These considerations are part of the daily practice found in patients with non-muscle invasive urothelial carcinoma in oncological follow-up.

2) Point 2: lack of enough information for the study cohort: how many diagnosis of cancer per patient? Previous intravescical therapies? Number and size of the lesions?

2.1) Response to point 2: We have carried out an extensive review of the patients included in the study and have included the following information:

Histopathological diagnosis of cancer per patient, therapy received, number of recurrences, clinical course, type of study during follow-up. In addition, we have been able to study the cytology of each patient, at the time of performing the methylation test.

3) Point 3: The greatest limitation is the lack of a reference after the tests? How do we know that either cytology or EpiCheck correctly discovered a tumor? How can we calculate sensitivity, specificity etc.?

3.1) Response to point 3: To try to resolve these doubts, and as we explained in the previous point, we have analyzed the studies after the result of the methylation test, mainly cystoscopy, some underwent randomized biopsies (BMN), or imaging studies. All these studies were criteria of the treating urologist. Based on these data, we implement statistical analysis in order to determine the positive predictive value, negative predictive value, sensitivity and specificity.

Finally, after the validation study in our center, it was decided to implement the methylation test in daily clinical practice, which has allowed a complementary test to be carried out in patients with urothelial carcinoma under follow-up.

Reviewer 2 Report

The authors presented the performance of the Bladder EpiCheck® methylation test at their institution. They demonstrated the usefulness of a DNA methylation test in the patients diagnosed with urothelial carcinoma. However, the usefulness of Bladder EpiCheck® was already published by other researchers, including Witjes et al.(Eur Urol Oncol 2018), Mancini et al. (Int J Mol Sci 2020) and Thomas et al. (Nat Rev Urol 2022). This manuscript has less new additional data, therefore, it has less impact to be published in the journal.

Author Response

Dear reviewer, thank you for your valuable comment and time. We have collected your comments in order to improve our work and so that it can be assessed for possible publication. Here are the changes made:

1) Point 1: The authors presented the performance of the Bladder EpiCheck® methylation test at their institution. They demonstrated the usefulness of a DNA methylation test in the patients diagnosed with urothelial carcinoma. However, the usefulness of Bladder EpiCheck® was already published by other researchers, including Witjes et al.(Eur Urol Oncol 2018), Mancini et al. (Int J Mol Sci 2020) and Thomas et al. (Nat Rev Urol 2022). This manuscript has less new additional data, therefore, it has less impact to be published in the journal.

1.1) Response to point 1: Dear reviewer, thank you for your valuable comments and time. We have collected your comments in order to improve our work and so that it can be assessed for possible publication. Here are the changes made:

There is no doubt that the usefulness of the methylation test has been reported in other studies, and as its negative predictive value has been investigated, it is the key element for its implementation in daily clinical practice. The main focus of the research is its ability to perform follow-up and diagnosis in patients with high-grade urothelial carcinoma, which are the ones with the most ominous prognosis for the patient. However, in daily pathological practice, the presence of morphological alterations in the context of treated patients sometimes makes it difficult to establish a conclusive diagnosis, in which atypia is a widely accepted category. In the case of urine cytology, since the implementation of the Paris system for reporting this type of material, diagnostic category III corresponds to atypical urothelial cells. All the studies that use the methylation test as mentioned above, focus on high-grade urothelial carcinoma and our goal is to investigate whether the test can be used as a complementary test in cases of atypical urothelial cells, since this diagnosis may require invasive diagnostic interventions, such as cystoscopy study with or without biopsy, according to the observer's expertise. Likewise, due to the multifocality of urothelial carcinoma throughout the excretory urinary system, alterations can be observed that mask the cytological findings and therefore require more extensive diagnostic tests.

Reviewer 3 Report

General comment

The manuscript entitled “Usefulness of the Urine Methylation Test (Bladder EpiCheck) in patients with Bladder Cancer and Cytological Diagnosis of Atypical Urothelial Cells. An Institutional Study” aims to evaluate the role of bladder Epicheck in the discrimination of atypical cells in cytology compared to this novel urine analysis in patients in follow up for NMIBC. Despite the manuscript could be interesting and provide real-life experience with this test, several concerns and limitations have to be addressed before considering the manuscript suitable for publication. In particular, the lack of a histopathologic analysis performed for patients who reported positive bladder epicheck test is the main limitation, in addition to the absence of clear inclusion and exclusion criteria. The statistical analysis is similarly lacking and the discussion is poor, repeating mostly concepts already reported in the introduction.

  • Major Issues

TITLE

The title should be revised accordingly to the content of the manuscript, it is not clear. There is no referral of bladder cancer patients in the manuscript (excluding the previous history)

ABSTRACT

The concept of bladder cancer diagnosed via cytology and cystoscopy is intrinsically wrong. The only certain diagnosis of bladder cancer and the difference between NMIBC and MIBC, is only made via histopathologic analysis of the specimen removed via TURBT. This is further reported in the introduction and should be revised and corrected.

INTRODUCTION

50-53: Before proceeding to cystoscopy? Clearly state the aim of your study and clarify the role of cystoscopy and cytology in bladder cancer.

MATERIALS AND METHODS

Inclusion and exclusion criteria have to be reported

93: Were the samples obtained at the same or different times?

109: Was performed no statistical analysis? Please explain.

RESULTS

116: and the current histopathological analysis? How did you confirm the presence/absence of bladder cancer?

A diagnostic post methylation test was not reported nor defined in the manuscript

DISCUSSION

The discussion is redundant and repeats concepts already cited in the introduction. I would use this section to compare your study with other similar studies and report thoroughly the role, the pros and cons of liquid biopsy. To this regard see also: https://doi.org/10.1002/cac2.12129 and https://doi.org/10.1016/j.critrevonc.2022.103577

Regarding the limitations, it has to be added the small sample size, the absence of histopathological confirmation and the retrospective nature

  • Minor Issues

INTRODUCTION

41-43: Also inflammatory states could limit the efficacy and reliability of urine cytology. Please add proper references.

MATERIALS AND METHODS

56-58: hematuria could similarly reduce the efficiency of bladder epicheck and urine cytology.

Author Response

Dear reviewer, we appreciate your time and your recommendations to improve our study. Please find below the modifications made, taking into account your important comments:

  • Major Issues

1) Point 1: TITLE

The title should be revised accordingly to the content of the manuscript, it is not clear. There is no referral of bladder cancer patients in the manuscript (excluding the previous history).

1.1) Response to point 1: Following your instructions, we have modified the title as described below, emphasizing that the selected patients are being followed up for non-muscle invasive urothelial carcinoma:

Usefulness of the Urine Methylation Test (Bladder EpiCheck®) in Follow-up Patients with Non Muscle invasive Bladder Cancer and Cytological Diagnosis of Atypical Urothelial Cells. An Institutional Study

2) Point 2: ABSTRACT

The concept of bladder cancer diagnosed via cytology and cystoscopy is intrinsically wrong. The only certain diagnosis of bladder cancer and the difference between NMIBC and MIBC, is only made via histopathologic analysis of the specimen removed via TURBT. This is further reported in the introduction and should be revised and corrected.

2.1) Response to point 2: As you correctly point out, the definitive diagnosis of urothelial carcinoma is through biopsy, which allows confirming the presence of cancer, in addition to providing the histopathological grade, in addition to its level of infiltration. We have corrected this point taking into account your correction, as follows:

Urothelial bladder cancer is one of the most common cancers worldwide and is a heterogeneous disease. Bladder cancer ranges from low-grade tumors that recur and require long-term invasive surveillance to high-grade tumors with high mortality. After the initial contemporary treatment in non-muscle invasive bladder cancer, recurrence and progression rates remain high. Follow-up of these patients involves the use of cystoscopies, cytology, and imaging of the upper urinary tract in selected patients. However, in this context, both cystoscopy and cytology show limitations.y. In follow-up bladder cancer the finding of urothelial cells with abnormal cytological characteristics is common. The main objective of our study was to evaluate the usefulness of a urine DNA meth-ylation test in patients with urothelial bladder cancer under follow-up and with a cytological finding of urothelial cell atypia. In addition, we analyze the relationship between the urine DNA methylation test, urine cytology and subsequent cystoscopic study. It is a prospective and descrip-tive cohort study conducted on patients with non-muscle invasive urothelial carcinoma between 1st January 2018 and 31st May 2022. A voided urine sample  and a DNA methylation test was performed from each patient. A total of 70 patients, 58 male and 12 female, with a median age of 70.03 years were studied. High grade urothelial carcinoma was the main hispathological diag-nosis. Of the cytologies, 41. 42% were cataloged as atypical urothelial cells. The DNA methyla-tion test was positive in 17 urine samples, 51 were negative and 2 invalid. We demonstrated the usefulness of a DNA methylation test in the follow-up of patients diagnosed with urothelial car-cinoma. The methylation test also helps to diagnose urothelial cell atypia.

3) Poin 3: INTRODUCTION

50-53: Before proceeding to cystoscopy? Clearly state the aim of your study and clarify the role of cystoscopy and cytology in bladder cancer.

3.1) Response to point 3: We have modified and clarified the objective of our study and clarified the role of cystoscopy and cytology in bladder cancer:

The main objective of our study was to evaluate the usefulness of a urine DNA methylation test in patients with urothelial bladder cancer under follow-up and with a cytological finding of urothelial cell atypia. In addition, we analyze the relationship between the urine DNA methylation test, urine cytology and subsequent cystoscopic study.

4) Point 4: MATERIALS AND METHODS

4.1) Inclusion and exclusion criteria have to be reported

4.1.1) Response to point 4.1: We have included inclusion and exclusion criteria:

The general inclusion criteria for patient selection were: age between 40-95 years, non-muscle invasive urothelial bladder cancer diagnosis, conventional treatments carried out under medical supervision, absence of severe psychiatric disorders, chronic alcoholism or drug addiction and adequate understanding of cystoscopy and adherence to follow-up standards. Patients who did not meet the inclusion criteria were excluded from the study.

4.2) 93: Were the samples obtained at the same or different times?

4.2.1) Response to point 4.2: We have clarified the question, since both the sample for the urine cytology and for the methylation test were taken at the same time.

Two urine samples were obtained from each patient simultaneously, one of which was processed for cytological study and the other for DNA methylation study. At the time of evaluating the results of both the cytology study and the methylation test, none of the investigators responsible for the analyzes were aware of the results of the tests.

4.3) 109: Was performed no statistical analysis? Please explain.

4.3.1) Response to point 4.3: We were able to perform statistical analysis in our study. Below we detail the methodology:

Statistical analyses were carried out in “R” (version 4.2.0) using “stats” library (version 4.2.0). To test the relationship between the epicheck result (positive or negative) and the diagnostic category of the cytology (I-V), Pearson’s Chi-squared statistical test was chosen. To assess the sensitivity and specificity of epicheck tests, cystoscopy results were considered as the reference standard and used to accordingly classify epicheck results. Given the invasiveness of cystoscopies and their impracticability for a screening study, Positive Predictive Value (PPV) was calculated including the prevalence of bladder cancer (1 in 27 in men, from www.cancer.org) as recommended by Molinaro 2015.

5) Point 5 RESULTS:

5.1) Poin 5.1: 116: and the current histopathological analysis? How did you confirm the presence/absence of bladder cancer?

5.1.1) Response to point 5.1: In our center, after the validation study for the DNA methylation test in urine, patients with a positive cytological diagnosis (category, IV and V) and/or positive methylation test, undergo cystoscopy and random biopsy of bladder (BMN).We have included this information in Table 1.  

5.2) Point 5.2: A diagnostic post methylation test was not reported nor defined in the manuscript

5.2) Response to Point 5.2.1: We have included the diagnostic information at our fingertips post-diagnosis of the methylation test.

6) Point 6 DISCUSSION:

6.1) Point 6.1: The discussion is redundant and repeats concepts already cited in the introduction. I would use this section to compare your study with other similar studies and report thoroughly the role, the pros and cons of liquid biopsy. To this regard see also: https://doi.org/10.1002/cac2.12129 and https://doi.org/10.1016/j.critrevonc.2022.103577

6.1.1) Response to point 6.1: Following your valuable comment we have made substantial changes to the discussion. We have focused these changes on the role of the bladder epicheck and its possible usefulness in routine pathology practice in cases diagnosed as urothelial cell atypia. We also explain the causes of urothelial atypia associated with the treatment evidenced in our patients.

6.2) Point 6.2: Regarding the limitations, it has to be added the small sample size, the absence of histopathological confirmation and the retrospective nature

 6.2.1) Response to point 6.2.: Following your comment we have included in the limitations the points indicated.

  • Minor Issues

7) Point 7 INTRODUCTION:

7.1) Point 7.1: 41-43: Also inflammatory states could limit the efficacy and reliability of urine cytology. Please add proper references.

7.1.1) Response to point 7.1: We have included the following references:

Cakir E, Kucuk U, Pala EE, Sezer O, Ekin RG, Cakmak O. Cytopathologic differ-ential diagnosis of low-grade urothelial carcinoma and reactive urothelial proliferation in bladder washings: a logistic regression analysis. APMIS. 2017 May;125(5):431-436. doi: 10.1111/apm.12657. Epub 2017 Feb 22. PMID: 28225151.

Sanfrancesco J, Jones JS, Hansel DE. Diagnostically challenging cases: what are atypia and dysplasia? Urol Clin North Am. 2013 May;40(2):281-93. doi: 10.1016/j.ucl.2013.01.006. Epub 2013 Feb 26. PMID: 23540785; PMCID: PMC5224879.

8) Point 8: MATERIALS AND METHODS

8.1) Point 8.1: 56-58: hematuria could similarly reduce the efficiency of bladder epicheck and urine cytology.

7.1.1) Response to point 7.1: We have included the following sentence: hematuria could similarly reduce the efficiency of bladder epicheck and urine cytology.